# The Italian Experience in the Development of Mesothelioma Registries: A Pathway for Other Countries to Address the Negative Legacy of Asbestos

**DOI:** 10.3390/ijerph20020936

**Published:** 2023-01-04

**Authors:** Corrado Magnani, Carolina Mensi, Alessandra Binazzi, Daniela Marsili, Federica Grosso, Juan Pablo Ramos-Bonilla, Daniela Ferrante, Enrica Migliore, Dario Mirabelli, Benedetto Terracini, Dario Consonni, Daniela Degiovanni, Michela Lia, María Fernanda Cely-García, Margarita Giraldo, Benjamin Lysaniuk, Pietro Comba, Alessandro Marinaccio

**Affiliations:** 1Department of Translational Medicine, University of Eastern Piedmont, 28100 Novara, Italy; 2Collegium Ramazzini, Bentivoglio, 40010 Modena, Italy; 3Occupational Health Unit, Fondazione IRCCS Ca’ Granda Ospedale Maggiore Policlinico, 20122 Milan, Italy; 4Department of Occupational and Environmental Medicine, Epidemiology and Hygiene, Italian Workers’ Compensation Authority, 00143 Rome, Italy; 5Department of Environment and Health, Istituto Superiore di Sanità, ISS (Italian National Institute of Health), 00161 Rome, Italy; 6Mesothelioma Unit, Azienda Ospedaliera SS Antonio e Biagio e Cesare Arrigo, 15121 Alessandria, Italy; 7Departamento de Ingeniería Civil y Ambiental, Universidad de Los Andes, Bogotá 111711, Colombia; 8Unit of Cancer Epidemiology, Regional Operating Center of Piemonte (COR Piemonte), University of Torino and CPO-Piemonte, 10126 Torin, Italy; 9Palliative Care Vitas, 15033 Casale Monferrato, Italy; 10IRD (MàD by CNRS), UMR Prodig, 93222 Aubervilliers, France

**Keywords:** mesothelioma, mesothelioma registry, asbestos

## Abstract

Asbestos (all forms, including chrysotile, crocidolite, amosite, tremolite, actinolite, and anthophyllite) is carcinogenic to humans and causally associated with mesothelioma and cancer of the lung, larynx, and ovary. It is one of the carcinogens most diffuse in the world, in workplaces, but also in the environment and is responsible for a very high global cancer burden. A large number of countries, mostly with high-income economies, has banned the use of asbestos which, however, is still widespread in low- and middle-income countries. It remains, thus, one of the most common occupational and environmental carcinogens worldwide. Italy issued an asbestos ban in 1992, following the dramatic observation of a large increase in mortality from mesothelioma and other asbestos-related diseases in exposed workers and also in subjects with non-occupational exposure. A mesothelioma registry was also organized and still monitors the occurrence of mesothelioma cases, conducting a case-by-case evaluation of asbestos exposure. In this report, we describe two Italian communities, Casale Monferrato and Broni, that faced an epidemic of mesothelioma resulting from the production of asbestos cement and the diffuse environmental exposure; we present the activity and results of the Italian mesothelioma registry (ReNaM), describe the risk-communication activities at the local and national level with a focus on international cooperation and also describe the interaction between mesothelioma registration and medical services specialized in mesothelioma diagnosis and treatment in an area at high risk of mesothelioma. Finally, we assess the potential application of the solutions and methods already developed in Italy in a city in Colombia with high mesothelioma incidence associated with the production of asbestos-cement materials and the presence of diffuse environmental asbestos pollution.

## 1. Foreword

Asbestos has been banned in 67 countries, most of them with high-income economies. The large-scale industrial use of asbestos includes in particular shipbuilding and repair, asbestos-cement manufacturing and production, railways, building construction, chemical industry, and many other industrial sectors. Now, most asbestos consumption and production is concentrated in low- and middle-income countries. The asbestos-induced tragedies experienced in high-income countries will eventually surface in communities located in low- and middle-income countries, as it is beginning to be documented in Osasco (Brazil) and Sibaté (Colombia). The similarities between what was observed decades ago in towns where asbestos plants operated in Europe and what is observed now in towns exposed more recently is striking and sad. History is repeating itself 20–30 years later. Moreover, the tragedies are more profound and absurd as we observe that the lessons on the hazards generated by asbestos use have not yet been applied in low- and middle-income countries. History is once again repeating itself as if we had not learned anything about the best strategies on how to deal with the negative legacy of asbestos use.

Among the wide array of asbestos related diseases, mesothelioma is emerging as the most insidious, as it also hits after environmental exposure, and no cure is possible. Trying to answer this conundrum, a group of researchers, physicians, and government officials compiled their extensive experiences in the strategies implemented in Italy since the 1980s, including the investigation of mesothelioma clusters in areas where asbestos-cement facilities have operated, the establishment of mesothelioma registries, and the treatment of patients diagnosed with mesothelioma. Some of the indications will also be useful in respect to the other asbestos-related diseases although this paper is limited to mesothelioma. These experiences are intended to guide other countries and communities affected by asbestos and by the health consequences of this hazardous material.

## 2. The Role of Mesothelioma Registration in Descriptive and Etiological Epidemiological Studies

### 2.1. Casale Monferrato: From the First Evidence of Mesothelioma to the Development of a Formal Population-Based Mesothelioma Registry in a High-Risk Area

Asbestos-cement production in Casale Monferrato started at the turn of last century, when the first Italian factory was set up in 1907 using the Hatschek production process, patented in 1902 and with the “Eternit” brand name. The factory grew quickly, with the increasing demand of cheap building materials [1]. In those days, Casale Monferrato was already an important production center for cement made out of local limestone. The raw asbestos was imported both from the Balangero mine located at ~150 km from Casale Monferrato and from abroad, with different countries of origin depending on the type of fiber used in the production process and the historical period of the acquisition (i.e., Canada, the USSR, and Australia). In the factory, raw asbestos and cement were mixed in the appropriate proportions to form a slurry that was used to prepare slates, tubes, high-pressure pipes, and other products using a process that was common to most asbestos-cement plants [2]. Crocidolite was part of the mixture in different proportions according to the product: for high-pressure pipes, asbestos could be up to one-third of the mixture [3,4].

The factory size was impressive, with an extension of more than 90.000 square meters. It was located in the western suburbs of the town, with the first residential areas at less than 500 m from the premises. The main Eternit products warehouse was located on the opposite eastern side of the township close to the railway station, and it included a large railway yard. Raw asbestos was moved from the railway yard to the factory, while finished products were shipped from the factory to the warehouse—in both directions, right across downtown Casale Monferrato. Initially, the main transport system of Eternit was a private narrow gauge railway; by the late 1950s, road transportation was preferred.

In the early 1970s, the period of the largest growth of the factory, the factory employed about 1500 workers. Approximately, 5000 employees worked at the factory between 1907 and 1986. In 1986, Casale Monferrato had a population of 40,000. The factory employed many women, especially during World War II. After a long period of prosperity, in the 1980s, the Italian asbestos-cement market collapsed, and the factory faced an economic crisis that eventually led to filing a petition in bankruptcy and to the cessation of activities in April 1986. The Eternit group at the time included five factories in Italy, of which the one in Casale Monferrato was the largest and oldest [3]. The other factories were located in the areas of Torino [5], Reggio Emilia [6], Naples [7], and Siracusa [8].

During most of the 20th century, in the area of Casale Monferrato, the local demand for asbestos-cement products was high, as can be expected, given that they were locally produced. In a survey carried out between 2001 and 2003, almost 800,000 square meters of asbestos-cement roofs were identified in the municipality of Casale Monferrato and its surrounding towns [9]. Moreover, the scrap asbestos-cement materials were handed out for free to citizens and were largely used to harden footpaths and courtyards or even as insulation materials in roofing [10,11]. In the city of Casale Monferrato, the use of asbestos cement materials was prohibited in 1987 with a mayoral decree after the publication of the first scientific evidence on mesothelioma in the area [12]. The remediation activities began in the early 1990s, after the national asbestos ban law [13] and under the strong pressure of Trade Unions and Victims’ Supporting Committees. Remediation plans were oriented to the proper disposal of friable asbestos, the removal of asbestos-cement products in public and private buildings, and the demolition of the Eternit facility, which was completed in 2010 [14]. Most of the planned remediation activities have now been completed.

The evidence of the health effects associated with asbestos was only partially known to the workers and to the population until the early 1980s. Until then, the current information on health risks was limited to asbestosis in workers. In 1983, a preliminary report in the form of a medical school thesis about a case series of mesothelioma cases in residents of the Casale Monferrato district highlighted, in addition to the large number of cases, the high proportion of women and of cases without occupational exposure [15]. After that initial evidence, the Regione Piemonte administration sponsored the development of formal epidemiological studies. In 1984, the first mortality cohort study of workers of the Eternit plant was launched, and results were published in 1987 [16]. It was followed by the first cohort study about the wives of asbestos-cement workers [17]. Both cohorts have been updated over the decades, and their findings have been regularly reported to the population. Previously, workers and citizens had never been informed about asbestos-related health risk by the factory establishment. Monitoring of asbestos exposure was very limited both in the factory [3] and in the urban environment [10], but results showed that pollution was impressive.

The cohort of Eternit workers in the most recent update [3] included 3434 blue-collar workers (2657 men and 777 women) who were active at the plant on 1 January 1950 or were hired between 1950 and 1986, accounting for over 85,000 person-years. One hundred and thirty-five deaths from pleural and 52 from peritoneal malignancy were observed in the follow-up period from 1965–2003. Corresponding standardized mortality ratios (SMRs) were 32.0 and 27.9 in men and 62.1 and 25.7 in women. Corresponding SMRs for lung cancer were 2.4 in men and 2.2 in women, based on 237 and 12 deaths, respectively. The cohort also showed a very large increase in deaths from asbestosis: 162 deaths in men and 24 in women, with less than one case expected. These risks were confirmed by the later study by Luberto et al. on a pool of Italian asbestos-cement workers cohorts, including the Casale Monferrato cohort [4].

The cohort of workers’ wives also showed a large increase of mesothelioma deaths and incident cases. Between 1965 and 2003, 21 deaths from pleural malignancy were observed vs. 1.2 expected, and the SMR for pleural cancer was 18.0 (95% CI 11.1–27.5). In the study period from 1990–2001, 11 incident cases of pleural malignant mesothelioma (MM) were observed by the Piedmont Mesothelioma Registry (SMR = 25.2; 95% CI, 12.6–45.1).

Taken together and compared with the tally of deaths from pleural malignancy in the area, these studies showed that the overall number of deaths attributed to occupational or domestic exposure to asbestos explained only 50% of all observed cases of pleural malignancies. This confirmed the preliminary observations of the local medical staff regarding the proportion of cases without direct occupational asbestos exposure and urged to establish a system for a population-based collection of pleural mesothelioma cases and analysis of their causes. With the help of a small grant from the International Agency for Research on Cancer (IARC), the collection of incident mesothelioma cases in residents of the Casale Monferrato district (approximate population of 100,000 inhabitants) was started. A survey was conducted for the period from 1980–1991, perusing the registries of the pathology units in the relevant hospitals of Regione Piemonte and the neighboring area of Pavia Province, as was customary at the time. The results indicated a very high incidence of pleural mesothelioma and confirmed the hypotheses relating the excess not only to occupational but also to domestic and environmental exposures [18,19]. A panel of pathologists evaluated the cases occurring from 1980–1989 and confirmed the diagnoses of cases [20,21]. This exercise contributed to the standardization of the criteria of diagnostic evaluation of MM cases [21].

These studies, overlapping between descriptive and analytical epidemiology, also prompted the implementation of a regional mesothelioma registry (Piedmont Mesothelioma Registry) monitoring the epidemic of MM in the whole region, including the Casale Monferrato area [22]. As for the most recent data (i.e., years 2013–2017), the incidence of pleural MM in the district of Casale Monferrato was 54/100,000 py in men (age-adjusted over the EU population, based on 132 cases) and 32/100,000 py in women (based on 97 cases) [23]. The Piedmont Mesothelioma Registry was one of the local registries contributing to the development of the Italian Mesothelioma Registry (Registro Nazionale Mesoteliomi: ReNaM), as described in a later section. 

The protocols implemented in the registry follow ReNaM guidelines [24] and include a personal interview of mesothelioma cases (or their next of kin), with a standardized collection of the lifelong opportunities for occupational, residential, environmental, and other types of exposure to asbestos. Definitions of residential and environmental exposure include the exposure from external sources, and the former is used when the exposure is linked to the location of the house; the latter has a more general meaning referring to general asbestos contamination of the environment. Domestic exposure (also defined as “home-related”) refers to the exposure from asbestos present in the house or in tools used for everyday life. Familial exposure is used when the exposure is linked to the occupational exposure of family members, most often because they carried home asbestos with soiled work clothes [24].

Collection of cases became systematic in 1990 although earlier cases were included whenever possible. Initially, the characterization of cases was retrospective, and later, it became concurrent. Interviews were conducted for all cases that offered the possibility of a direct or proxy interview [25]. 

The mesothelioma registry has set the basis for further analytical studies on mesothelioma etiology, including both case control analyses [10,26,27,28] and the analysis of mesothelioma incidence in the cohorts of asbestos workers and wives [3,29,30]. In the case control studies, asbestos exposure was carefully evaluated, including both occupational and non-occupational exposure, and residential history was geocoded. The results confirmed the dose–response relationship of mesothelioma and asbestos exposure and showed the spatial distribution of mesothelioma risk in the area of Casale Monferrato [10,27,28,31]. The sources of environmental exposure include the direct emissions from the plant, the transport of raw asbestos and of asbestos cement products to and from the factory, and the use of asbestos cement tailings in the city [10,27,31]. Methodologically the activity confirmed the importance of direct interviews whenever possible, as they are more informative on asbestos exposure than proxy interviews. 

The data from the Piedmont Mesothelioma Registry documented the relative contribution of occupational and non-occupational asbestos exposure in Casale Monferrato. Table 1 reports absolute numbers of incident mesothelioma cases in the population of Casale Monferrato broken down by sex, period of diagnosis, and type of exposure. The table presents the number of mesothelioma cases diagnosed in Casale Monferrato population, either ever employed in Eternit premises (Eternit employees or hired by subcontracted enterprises in charge of activities such as cleaning, transportation, etc.) or reporting indirect exposure to asbestos from the Eternit facility. For the present analyses, the latter category included the following: (i) residents ever living within 2 km from the Eternit plant; (ii) spouses and cohabitants of Eternit workers; and (iii) subjects ever living in houses where asbestos materials originating from the Eternit plant (such as scrap asbestos-cement materials, including the fine dust from lathing asbestos-cement pipes) were present. These figures are not exhaustive of the number of cases associated to Casale Monferrato and to Eternit: in residents of other towns, 66 mesothelioma cases were diagnosed as related to past employment at Eternit (16 women and 50 men) and 168 with past environmental exposure from living in Casale Monferrato (78 women and 90 men) (data not tabulated). It is important to highlight that the number of environmentally exposed cases have declined less compared to occupationally exposed cases, which has resulted in an increased environmental/occupational ratio for mesothelioma cases. For a large proportion of subjects, the different sources of non-occupational exposure, in particular domestic, familial, and environmental, showed some overlap.

The different gender ratio between the mesothelioma cases associated with occupational and non-occupational asbestos exposure is also noteworthy. The point will be further addressed in the following section on the ReNaM.

A further noteworthy result was represented by the analyses of the relationship between asbestos and genetics in the etiology of mesothelioma. These studies too often are conducted on clinical case series that tend to overestimate the occurrence of genetic trait syndromes, as happened for the first results on BAP1 syndrome [32]. The availability and use of a reliable, population-based registry of mesothelioma cases made it possible to estimate accurately the etiological role of the BAP1 mutations, which explain the rare syndromic cases of mesothelioma but have a very small role in the sporadic cases [33]. Additional analyses of the registry-based cases led to the estimation of the prevalence of cases carriers of driver genetic mutations. It was established that these cases were more sensitive to exposure to asbestos, consistent with the multi-stage etiology of cancer [34,35].

### 2.2. Case Study: Mesothelioma Occurrence in Broni—The Role of the Mesothelioma Registry

Broni is a small town (about 9600 inhabitants) in the Province of Pavia, Lombardy region, northwestern Italy, where a factory named Fibronit produced asbestos-cement products from 1932 to 1993. The Fibronit was the second oldest and largest (in terms of worker-years) asbestos cement factory in Italy (after the Eternit in Casale Monferrato) [4,36,37]. As of the 1970s, several tasks were performed manually, and there were no ventilation systems, workers used no personal protection devices, and in general, there were poor work hygiene practices [36]. Only between the end of the 1970s and the 1980s, measures were taken to reduce airborne fiber concentrations [36]. The use of asbestos was stopped in 1992–1993. The factory continued the production of cement products until 1997; however, no remediation works were undertaken, and the plant definitely closed in 2000. In the following years, remediation works began in the factory area and in other parts of the town of Broni and are still ongoing. In 2002, the perimeter of the factory was included by law 388/2000 in a government list of environmentally contaminated sites (Siti di Interesse Nazionale, SIN) in Italy [8,38].

Since the 1990s, several studies documented high pleural mesothelioma mortality and incidence among Fibronit workers and Broni area residents [39,40,41,42,43,44,45]. In particular, the Lombardy Mesothelioma Registry (Registro Mesoteliomi Lombardia, RML) documented that in the period from 2000–2012 in Broni, the crude mesothelioma incidence rate (per 100,000 person-years) was 100.0 in men and 68.4 in women, among the highest in Italy. Corresponding incidence rates in the whole Lombardy Region were 4.7 (men) and 2.5 (women) [42].

Notwithstanding the analogies with the Casale Monferrato case, Broni did not receive, for various reasons, the same national and worldwide resonance. One of the reasons is certainly the small size of the town (about one-quarter compared to Casale Monferrato), so the absolute number of people with asbestos-related diseases is smaller than in Casale Monferrato.

A more important factor is probably the different response of the two communities to the asbestos tragedy. The parallel histories of Casale Monferrato and Broni, with their similarities and differences, have been recently analyzed from the historical and social points of view [46]. In Casale Monferrato, thanks to the initial efforts of a few members of the trade unions, gradually including larger numbers of Eternit workers and of their families and finally of the whole population, the problem was brought up to the general attention, and initial research was supported by the local and regional administration. Over the years, the whole community, with the involvement of the local association of asbestos victims (AFeVa), took care of the problem in all medical, legal, and environmental aspects. Finally, the case of Casale Monferrato appeared on the political scene at a national level. On the contrary, in Broni, an opposite effect took place, namely a sort of removal of the problem, in which probably also the dilemma of choosing between a safe environment and employment had a role.

The RML started data collection in 2000 as a part of the ReNaM [47]. Data collection procedures follow the ReNaM guidelines [24]. In summary, it collects all MM cases diagnosed in people living in Lombardy (currently almost 10 million inhabitants) at the time of diagnosis from regional and extra-regional hospitals. The final clinical diagnosis is reviewed on an individual basis after examining the relevant clinical information, including imaging and pathology reports. Exposure information is collected through the ReNaM questionnaire administered to patients or next of kin by trained interviewers.

The information stored in the RML database allows the tracking of clusters of MM cases attributable to the same workplace, including Fibronit workers and subjects who had ever lived in Broni. Using these data, the assessment of the impact of the Fibronit factory on MM incidence in the area of Broni was updated through the year 2016 [48]. Overall, in the period from 2000–2016, we identified 218 subjects (standardized incidence ratio–SIR: 8.4) with MM (207 pleural, 11 peritoneal) in the Broni area with asbestos exposure related to the Fibronit asbestos-cement factory. There were 97 men (SIR: 6.4) and 121 women (SIR: 11.1). In plain words, compared with the regional average, mesothelioma was 6 times greater in men and 11 times in women. Regarding exposure source, we observed the following:*Occupational exposure in the Fibronit factory:* 56 MM cases—49 men (41 pleural, 8 peritoneal; SIR: 20.6) and 7 women (5 pleural, 2 peritoneal; SIR: 50.0);*Familial exposure:* 39 cases who had been cohabitants of Fibronit workers—10 men (all pleural MM; SIR: 3.8) and 29 women (28 pleural, 1 peritoneal; SIR: 17.7);*Environmental exposure:* 123 cases—38 men and 85 women—all with pleural MM. Of these, 91 subjects had ever lived in Broni—31 men (SIR: 8.1) and 60 women (SIR: 16.7). Moreover, there were 25 subjects with MM residing in Stradella (an adjacent town) at the time of diagnosis—6 men (SIR: 3.8) and 19 women (SIR: 13.0).

In summary, thanks to the registry, we documented a high impact of past asbestos exposure from the asbestos-cement factory Fibronit on workers, their families, and residents in the towns of Broni and Stradella, with a total of approximately 194 excess cases estimated in a 17-year period (211 observed cases against ~17 expected).

The absolute impact from occupational exposure was larger in men, while MM burden from familial and environmental exposure was higher in women. Note that these gender differences are in part dependent on the hierarchical classification scheme used: in fact, subjects with both occupational and non-occupational asbestos exposure (mostly men) are assigned to occupational exposure. The fact that three-fourths of cases were attributable to familial or environmental exposure highlights the importance of a global assessment of the health effects of asbestos exposure in the community at large. Based on these results, in 2021, the Environment Department of Lombardy Region submitted to the Ministry of the Environment the request to extend the SIN to entire town of Broni. This will facilitate the asbestos remediation as occurred in Casale Monferrato.

## 3. Mesothelioma Registration: The Italian National Mesothelioma Registry (ReNaM) and International Experiences

Italy is one of the countries most involved in the epidemic of asbestos-related diseases due to the large use of asbestos in the past and the number of exposed subjects among workers and general population until the asbestos ban issued in 1992 [13]. About 3,748,550 tons of raw asbestos have been used in Italy in a large spectrum of industrial activities, with a peak between 1976 and 1980 (more than 160,000 tons/year in this period) [47]. According to the long latency of asbestos-related diseases, which for MM could be more than 40 years, Italy is now experiencing severe public health consequences. It is necessary to take into account that mesothelioma is strictly a dose-dependent disease with no exposure threshold; therefore, the risk of disease is greater after the more important exposure periods, but it cannot be excluded even after a short, discontinuous, and not particularly heavy exposure. In this context, a permanent and mandatory epidemiologic surveillance system of MM is active based on a national registry (ReNaM) established by the Italian national authority for workers’ compensation (Inail) with force of law [24]. The ReNaM aims are to provide estimates of MM incidence at national population level, to assess and record asbestos exposures of collected cases, and to identify any possible underestimated or unknown source of asbestos contamination. ReNaM represents one of the few population based mesothelioma registries in the world [49].

ReNaM is an epidemiological surveillance system organized as a network of regional operating centers (Centri Operativi Regionali in Italian—COR) that have gradually been established in all 20 Italian regions. Italian regions did not join the ReNaM network simultaneously, and some regions started registration before the formal setup of ReNaM, providing a substantial basis for its organization. Recently, the ReNaM network published the Seventh ReNaM National Report, which includes 31,572 incident MM cases, collected in the period from 1993–2018. For 24,864 of them, the modalities of asbestos exposure were investigated, finding an occupational asbestos exposure in 69.1% of interviewed patients, familial exposure in 5.1%, environmental exposure in 4.3%, and leisure-related exposure in 1.5%. The exposure to asbestos is unknown or unlikely in 20% of interviewed subjects [50].

The CORs actively search and register incident cases of MM from healthcare services that diagnose and treat cases (especially pathology and histology units and pulmonology and chest surgery wards). At COR level, expert physicians classify MM cases in three categories, according to the level of diagnostic certainty [24].

Occupational history, lifestyle habits, and residential history are investigated using a standardized questionnaire [24]. The CORs periodically transmit data to ReNaM, which provides epidemiological analyses, publishes national reports, and promotes specific research projects.

The Italian experience of MM incident cases registration provides evidence about the capacity of an epidemiological surveillance system to contribute to scientific knowledge, to increase the effectiveness of the insurance system for occupational diseases, to support the asbestos exposure prevention policies, and to gain awareness of health effects risks in exposed workers and general population. The median survival period for pleural MM is still very poor and is shorter for peritoneal and pericardial mesothelioma [51,52,53]. Age, site, and histological type have been identified as significant prognostic factors (young age and epithelioid morphology seem to be favorable prognostic factors). The estimation of the latency period is quite complex in subjects with severe health conditions and a remote occupational experience to investigate, mainly because it is extremely difficult to establish the exact onset of the asbestos exposure. Recently, the limits of the estimation of median latency in cross-sectional cases provided in a population-based observational study were repeatedly underlined [54]. In ReNaM patients, estimated median latency was longer than 40 years from the beginning of exposure [55]. In the framework of ReNaM research production, the trend of mesothelioma occurrence by asbestos consumption in Italy was described for the first time in Italy. According to asbestos consumption in the past and using a set of age-period-cohort models, the forecast scenarios of mesothelioma mortality was estimated, predicting the plateau of the disease incidence and mortality rates between 2015 and 2025 [56,57,58,59]. The geographical distribution of mesothelioma cases has been used to identify geographical clusters [60,61]. Mesothelioma occurrence was used as a proxy for asbestos exposure in order to estimate the extent of asbestos-related lung cancer mortality, finding a ratio of 1:1 between the two diseases [62]. According to the purposes fixed by law, ReNaM carries out the surveillance system of mesothelioma cases of all anatomical sites. Detailed analyses of peritoneal mesothelioma cases’ epidemiology have been performed comparing incidence and mortality data [63]. Recently, ReNaM, with the external collaboration of the Universities of Bologna and of Eastern Piedmont, produced the first analytical study on the association between asbestos exposure and the risk of pericardial and tunica vaginalis testis mesothelioma [64]. It is necessary to underline that only in an epidemiological surveillance system with a huge dimension of the observation period (more than 1000 million person/years) could a case-control study on these rare diseases be planned and carried out. The economic sectors involved in asbestos exposure include the wide spectrum of industries in the well-known traditional sectors, such as shipbuilding and asbestos-cement industries, but also unexpected sources of contamination [65,66]. It is indubitable that the identification of relevant industrial sectors, in particularly when exposure is unexpected, is a topic issue for public health policies and for the effectiveness of insurance and welfare systems. ReNaM has shown asbestos exposure in unexpected and unknown contexts such as wine production, jewelry, dental technicians, theaters, art work, and many others. In this context, an analysis of the relationship between the epidemiological surveillance system findings and compensation claims frequency has revealed the wide geographical variability, suggesting the need to strengthen the awareness of the occupational origin of the disease through specific formation plans [67]. Recently, the psychological distress and care needs of mesothelioma patients and asbestos-exposed subjects were investigated in the ReNaM network, suggesting that MM is associated with high levels of psychological distress, despair, and mental health difficulties for patients and caregivers [68]. Environmental exposure to asbestos is a crucial topic in ReNaM research activities. Exposure to asbestos due to the residence, to cohabitation with exposed subjects, or to leisure activities has been repeatedly analyzed in ReNaM network, documenting that 10.2% of mesothelioma cases are due to non-occupational exposure to asbestos [69,70]. The most significant source of risk is residence near an asbestos-cement plant. It is necessary to define policies and strategies for increasing prevention tools and for dealing with compensation rights for MM cases induced by non-occupational exposure to asbestos. In cooperation with the Istituto Superiore di Sanità, ReNaM has evaluated the occurrence of mesothelioma in the “contaminated sites”, as defined with force of law [38]. The findings have showed clear mesothelioma excesses not only in sites where asbestos is reported as a source of contamination but also in areas labeled for environmental remediation due to other sources of pollution [38,71]. This evidence confirms the wide range of working and living environments affected by asbestos exposure. Despite the interest for public health policies, studies on the economic costs of asbestos-related diseases for society are limited. Using an econometric model, an estimate of EUR 33,000 per patient for medical care costs and EUR 25,000 for insurance and compensation have been calculated in ReNaM context [72].

Italy presents a large presence of women among mesothelioma cases due to the relevance of non-occupational exposures and to the historically high female workforce participation in several industrial settings involving asbestos exposure, including the non-asbestos textile sector. ReNaM has discussed how the awareness of occupational or environmental origin of mesothelioma in women could improve the efficiency of the public compensation system and the prevention policies, redefining the tools for investigating asbestos exposure from a gender perspective. Overall, the gender ratio (M/F) is 2.6 in the ReNaM network [50]. Similar observations have been presented elsewhere: in Australia, the incidence rate in 2017–2020 was 4.2 in men and 0.9 in women [73], whereas the French National Mesothelioma Surveillance Programme (PNSM), currently recording incident MM cases in 26 districts, accounting for about one-quarter of the French population, provides evidence of a gender ratio equal to 3.7 [74]. However, the magnitude of mesothelioma incidence in women is positively correlated with the intensity of environmental exposure and with the dimension of female workforce in economic sectors traditionally involved in asbestos exposure (mainly the non-asbestos textile sector) [75]. The analysis of mesothelioma cases in Casale Monferrato (Table 1) showed a different gender ratio in relation to occupational or non-occupational exposure.

The economic sectors involved in asbestos exposure in the Italian industrial context have been repeatedly investigated. The “Constructions” sector is very large and appeared as the sector with the highest number of exposed cases although it is not the sector at the highest risk. ReNaM researchers have underlined that there is a need to implement education and training for workers involved in activities such as remedial, maintenance, and building renovations, especially with reference to old buildings, for avoiding the risk of exposure to asbestos that remains a real concern [76].

Worldwide, specific surveillance systems of incident MM cases, including anamnestic individual analysis, are scarce [45,49]. To the best of our knowledge, only a few are comparable for information completeness, environmental exposure assessment, and territorial coverage and are currently active. Australia, France, and South Korea registries are based on active search of MM cases, and the anamnestic exposure to asbestos is measured through a standardized questionnaire. In other countries (the Netherlands, Germany, Nordic countries, Australia, and New Zealand), MM cases notifications rely on general cancer registries and/or other systems (e.g., compensation requests for occupational diseases or spontaneous reports) or on deaths recording systems (UK) [45]. Recently, in the USA, the National Institute for Occupational Safety and Health (NIOSH), within the Centers for Disease Control and Prevention (CDC), announced the opening of a docket to obtain information on the feasibility of an incident MM cases registry [77]. The Italian experience (such as other similar nationwide experiences) provides evidence that an epidemiological surveillance system of MM incident cases, with individual anamnestic analysis of the modalities of exposure to asbestos, is a fundamental tool for producing scientific knowledge, for supporting and promoting the effectiveness of insurance and welfare systems, and for contributing to the asbestos exposure prevention policies.

In the Colombian context, the development of a systematic collection of data about mesothelioma incident cases at local level and progressively at nationwide level could significantly improve the awareness about asbestos-related health effects and contribute to the identification of unknown sources of contamination. The evaluation of the effects of measures for counteracting asbestos exposure could benefit from mesothelioma registration. Finally, the concrete experience of many Western countries (including Italy) shows the correct path to follow for contrasting asbestos exposure and the adverse health effects. This path, as well as an asbestos ban, includes the production of epidemiological evidence about asbestos-related health effects in a local community; the development of awareness in workers, people, and authorities about public health risks of asbestos use; and the production of evidence about the social, economic, and health advantages of complete asbestos removal.

## 4. Communication of Scientific Findings to Affected Communities and Stakeholders

The main steps for fighting asbestos-related diseases are the ban of asbestos use and the removal of asbestos in all places, which require actions at political and societal levels. The aim of this section is to show the role and the importance of communicating scientific findings on asbestos risk and health-related impact to the asbestos-contaminated communities and to the other stakeholders, with the purpose of contributing to increase community awareness and collective resilience. Recent experiences with asbestos communication are reviewed to present a communication framework to be shared and used in the Sibatè and other asbestos-contaminated areas in Colombia.

The international literature contains several directions emphasizing the relevance of communicating scientific evidence and results of asbestos risk and health-related effects. The World and Health Organization (WHO) has started disseminating scientific evidence on the carcinogenetic effects of asbestos through the IARC (International Agency for Research on Cancer) Monographs, with the first Monograph evaluating asbestos published in 1973 [78] and the WHO multi-lingual materials concerning asbestos disease prevention [79,80]. In the WHO European Region, the 53 member states have shared and renewed their commitment to pursue until achievement, develop, and implement their national programs for the elimination of asbestos-related diseases. This includes training and evidence-based communication aimed at strengthening capacities of health and environment professionals as well as formal and informal education for public’s understanding (point n. 13 of the Ostrava Declaration of the Sixth Ministerial Conferences on Environment and Health, 2017) [81].

Scientific evidence of the impact of hazardous substances exposure and related diseases, including asbestos occurring in both occupational and living environments, has to be communicated to the contaminated, affected communities [82,83]. The risk of asbestos exposure in living environments affects residents in areas characterized by asbestos exposure, such as because of asbestos-cement production and of uncontrolled asbestos waste disposal, as well as the relatives of asbestos workers (domestic exposure). Communication must be evidence-based and accessible to the involved communities and policy makers. This action is also necessary in those countries, including Italy, where asbestos has been banned for some decades [13], but pleural mesothelioma mortality excess is still registered in asbestos-affected municipalities and areas. In fact, asbestos health impacts are nowadays coming to light due to the past exposures [47,84,85] but also emerge in productive sectors considered non-traditional settings for asbestos exposure [66]. The lesson is that it is a long road to be free of asbestos [86]. Communication of health impacts of asbestos exposure is also critical in those countries, such as Colombia, that have just recently adopted the national asbestos ban. In Colombia, asbestos was banned in 2019 [87], and this action might have been favored by scientific evidence produced in the country itself, such as the first environmental and health studies on asbestos-related diseases in the Sibaté area [88].

The Istituto Superiore di Sanità (ISS, Italy) is collaborating with Italian and Latin American universities and health authorities to promote scientific cooperation frameworks including research, capacity building, training, and communication on asbestos and related diseases prevention [89]. Co-authored scientific and technical multi-lingual publications (Spanish, English, and Italian) have addressed the issue of asbestos risk and health-related impact with the goal of making evidence-based information and best practices accessible to key stakeholders in the involved countries [90,91,92]. Moreover, within the Italian national Asbestos Project (2013–2015), scientific cooperation was dedicated to exchange experiences with institutions of Latin American countries where asbestos was not yet banned. The aim was to foster environmental health literacy on asbestos as a relevant contribution to address environmental health and socio-economic aspects of asbestos risks and impacts in local contexts [93,94,95].

Starting from 2011, the first Italian–Colombian scientific collaborations were focused on asbestos issues. In 2017, the Istituto Superiore di Sanità and the Universidad de Los Andes signed a Memorandum of Understanding to formalize the cooperation framework in order to share research and practices on asbestos prevention. Researchers from other Italian public institutions participate in the cooperation initiatives. Collaboration activities have been dedicated to address priorities and local needs identified by the Colombian partners in detecting the asbestos health impact and in promoting prevention. These activities include epidemiological studies and the communication of scientific evidence on asbestos risk and of the findings of collaborative studies to affected communities [96].

Involving affected communities in communication is particularly relevant after the results of local surveys and epidemiological studies. In fact, it encourages the diffusion of new knowledge among stakeholders (health authorities, health professionals, public health students and researchers, trade unions’ representatives and NGOs, and citizens) on the severity of the health impacts of asbestos exposure in their occupational, domestic, and living environments. Epidemiological studies are essential for analyzing the distribution and for identifying clusters of asbestos-related diseases, in particular mesothelioma, a recognized marker of past asbestos exposures. MM mortality and incidence data should be communicated in a transparent way and by adopting a lay language in order to increase awareness and capacities at both individual and collective levels. Researchers involved in the studies have to adopt an ethical communication approach by sharing the scientific findings with national and local health authorities and other involved stakeholders in the affected community, including asbestos victims’ associations and citizens’ committees [97].

Evidence-based communication represents an essential contribution for increasing asbestos literacy and awareness in affected communities and policy makers, provides public health recommendations for the reduction of asbestos exposures in both occupational and living environments, and pushes for intervention priorities, including the implementation of environmental remediation actions and health prevention. This aspect requires the adoption of a cross-disciplinary approach ensuring the engagement of multidisciplinary skills from health and environmental sciences to social sciences [98]. It also requires the building of a communication process relying on a circular flow of evidence-based information among the researchers communicating the results of their studies, the affected population providing information on their asbestos exposure in occupational and living environments, and the institutional stakeholders in charge of asbestos prevention initiatives and related actions. Social actors, such as local associations of exposed workers, of asbestos victims’ relatives, and trade unions of asbestos workers, may play a relevant role in the communication process, fostering a network of relationships and bidirectional communications with involved researchers [99]. The consolidated experience on asbestos prevention in Italy, relying on increasing awareness, literacy, and multi-sectoral actions involving relevant stakeholders, has supported the Colombian efforts and progress in asbestos prevention, including the project of a mesothelioma registry for epidemiological surveillance.

Communication concerning mesothelioma and related evidence-based information should include at least the following notions:−Mesothelioma pathology, etiology, latency, and lethality of this disease;−Dose–response relationship between asbestos exposure and mesothelioma occurrence;−Protective effect of reducing asbestos exposure.

These concepts have to be properly explained using a lay and accurate language, which has to be understandable and usable by a non-expert audience as well as sensitive to the literacy and concerns of the local social actors. The socio-cultural dimension of a community acts as a mediator in communication processes, affecting the individual and collective perceptions of asbestos risk and health-related impact in affected communities [100]. 

Structured communication plans allow local affected communities and other stakeholders to better understand the aims and results of scientific investigations on the health and environmental and social impacts of asbestos use in their territories. Moreover, a validated epidemiological surveillance system is essential to identify and register the mesothelioma cases that occurred in a given period and monitor the new cases in the community. Awareness about the implementation and availability of a mesothelioma registry as well as understanding the meaning of mesothelioma incidence are crucial to appreciate and share the benefits of a structured epidemiological surveillance system for public health purposes.

The analysis of different communication experiences in Italy in several asbestos-affected communities shows differences in the effectiveness and impact of these initiatives, which also depends on the long-lasting relationships established among researchers, affected communities, and local authorities. This suggests that institutional, socio-economic, and cultural specificities of different local contexts may explain the different level of knowledge, awareness, and engagement of affected communities in the governance regarding the asbestos risk and impacts. The Istituto Superiore di Sanità coordinated a multi-institutional investigation on communication of the results of epidemiological studies and scientific findings in different asbestos-affected municipalities in Italy [100]. These initiatives were undertaken with the goal of increasing awareness of the health impact of asbestos exposures and community resilience. The Casale Monferrato experience can be considered as a best practice for the adoption of a circular communication process involving the community, the local authorities, and the researchers. Casale Monferrato represents a successful experience for creating a diffuse awareness and a social capacity building characterized by inclusiveness and participation of institutional actors (researchers, local administrators, health services, and educational system) and social actors (trade unions, associations of ex-exposed workers, and the associations of relatives of the asbestos victims). Institutional authorities and social actors are maintaining effective relationships with the involved researchers, confirming their participation in communication initiatives, such as public meetings, workshops, and initiatives, with representatives of other asbestos-affected communities visiting Casale Monferrato. The collaborative and participative approach shown by the local media further strengthens the relationships between scientists and the affected community [92]. Supporting training activities (such as “training the trainers”) have also been addressed to local students for their peer-to-peer educational initiatives [101].

In this framework, the epidemiological and health-related surveillance systems in Casale Monferrato are today still appropriate tools for increasing community resilience [102]. Addressing the communication needs of the most vulnerable population subgroups has supported their motivational capacities and emotional self-control in the circular communication process.

The psychosocial aspects, accompanying the clinical manifestation of the asbestos related-diseases and in particular the occurrence of MM, have to be considered in the elaboration of targeted contents of communication [68]. In this perspective, psychologists, physicians, and healthcare professionals play a key role in the dialogue with the patients and their family members through direct interactions and focus group therapies [103,104,105]. The discussion of this specific issue is outside the scopes of this section.

The Casale Monferrato experience demonstrates the importance of establishing and maintaining a long-lasting participative communication process since it is a long road to be free of asbestos. This is particularly true in those areas where the asbestos risks and related impacts are only recently emerging, such as the Sibaté municipality [88]. The need to elaborate on and adopt a suitable communication language is further corroborated by the involvement of institutional and social actors as well as by fostering their engagement for a long period of time [106]. Building a circular communication process is vital to preserve awareness and trust among the communities and key stakeholders, relying on transparency, perceived competence, and accountability, particularly for undertaking appropriate health prevention actions and environmental remediation interventions by the local authorities.

A structured scientific cooperation framework foreseeing the adoption of consolidated protocols for epidemiological studies design and prevention technologies such as epidemiological surveillance systems [91,107] can be further strengthened by the implementation of a structured and participative communication plan tailored to asbestos-affected communities [99].

## 5. The Mesothelioma Registry as a Tool for Clinical Approach and Treatment of Cases

The Clinical Mesothelioma Unit has operated since 2014 across the two hospitals of the area (Alessandria and Casale Monferrato). The unit is a recognized national reference center. A multidisciplinary team takes care of patients according to predefined diagnostic and therapeutic guidelines based on the most up-to-date scientific evidence and adapted to the local facilities. The medical staff visits about 140 newly diagnosed mesothelioma patients per year, and approximately 200 patients are in active follow-up. Patients most often come from the Alessandria province (including Casale Monferrato), but several patients are admitted from all over the Italian territory for second-opinion visits or willingness to receive experimental drugs. The unit takes active part in several national and international multicenter clinical trials, often partnering among the top recruiting centers worldwide. Mesothelioma peculiarities, such as rarity, dismal prognosis, and limited therapeutic options, require intergroup networking and collaboration across all the European reference centers. The aim is to gather precious data, to promote clinical and translational research, and to improve the knowledge about the underlying biological mechanisms. Hopefully, this will bring new drugs and/or diagnostic strategies in a setting that has been for years neglected by physicians and the pharmaceutical industry.

A multidisciplinary team composed by specialists from the different relevant disciplines manages the unit, including simultaneous and palliative care and psychological support. The team has a specific expertise in mesothelioma and is responsible for each patient along the entire process of diagnosis and treatment. A global care patient-centered model is adopted, based on the current clinical practice guidelines adapted to the local available facilities, within a structured disease-specific pathway.

As a mesothelioma reference center, recently, researchers successfully applied for membership in the European reference network, ERN EURACAN, a virtual network covering all rare adult solid tumors and connecting patients and healthcare providers across Europe. The aim of the network is to share expertise and improve access to care for patients affected by rare cancers, whose treatments require highly specialized knowledge and resources in order to reach the expected results and avoid negative consequences for their quality of life and outcome.

The key point of the mesothelioma unit is to maintain a very strong connection between research and whole patient care. In this view, in the last 12 years, 800 patients have been enrolled in clinical trials, and 250 have received experimental drugs within about 15 interventional multicenter studies. Research activity is increasing more and more, with active recruitment in a worldwide, competitive manner in order to ensure patients the best treatment options and the opportunity to have access to new drugs [108,109].

As a starting point to implement translational research projects, a Biobank has been setup at Alessandria Hospital since 1990. Now, it includes more than 800 biological samples, including about 500 mesothelioma samples, matched to clinical information from our database. The biobank has been recently accepted as part of BBMRI-ERIC, a European research infrastructure for biobanking. The Buzzi Onlus Foundation has supported the Biobank project financially. The foundation is aimed at the integration and networking of health care givers with asbestos victims and their families.

Research and clinical projects have been implemented in the last years thanks to the financial support by AFeVA (Asbestos Victims’ Patients Association), in collaboration with Eastern Piedmont and Turin Universities and with external research institutes, such as Mario Negri Pharmacological Research Institute in Milano. The MATCH project (Mesothelioma And Translational Clinical and Health-related data) is an observational and translational study aimed to study specific molecular signatures causing mesothelioma and ultimately to develop new diagnostic and therapeutic strategies, hopefully with the implementation of new target therapies. A first retrospective cohort was recruited and is currently under investigation to evaluate specific molecular biomarkers in correlation to patients’ outcomes (e.g., overall survival and response to treatments) as candidate predictive and/or prognostic factors [110]. The HERMES project—Hereditary Risk in MESothelioma—is an epidemiological study aiming to study the frequency and the impact of genetic alterations responsible for mesothelial cells transformation in a population of highly asbestos-exposed individuals in the Casale Monferrato area. The project has the objective to study pleural mesothelioma cell lines obtained from such a peculiar mesothelioma population (high asbestos exposure) and eventually aims at devising specific therapies for the cases with genetic alterations [111].

Finally, a regional asbestos research center dedicated to the research, surveillance, and prevention of asbestos risk has been located in Casale Monferrato since 2008 (Piedmont Regional specific decree). In particular, the center has the objective of epidemiological surveillance, research support aiming to optimize the diagnosis and the treatments of the asbestos-related diseases, and also to disseminate information and promote educational activities. In view of a more-and-more comprehensive management of all the asbestos-related and other environmental diseases, the Alessandria Hospital is preparing an application for the National Research and Health Care Institute award (Istituto di ricovero e cura a carattere scientifico—IRCCS).

The peculiar mesothelioma incidence pattern (clusterization), the high social and psychological impact, the dismal prognosis (median overall survival of about one year), and the current lack of effective therapeutic options make this disease one of the unmet-needs cancers [112]. There is a need to promote and support collaboration among the national and international reference centers (specifically with mesothelioma expertise) through data collection, which is crucial to share precious information. In this frame, the unit recently entered the EHDEN (European Health Data and Evidence Network) community, a European network aiming at harmonizing source data to the OMOP common data model locally. A common national or European multicenter database with fast access to patient data would provide an integrated and shared mechanism to enable the timely implementation of clinical trials and robust translational research.

As to our knowledge, there is no screening program for mesothelioma active in any country in the world since there is no screening test with adequate sensitivity and specificity, and above all, there is no curative treatment even for subjects diagnosed with in situ or early-stage mesothelioma. However, the emerging availability of drugs able to inhibit mesothelioma formation in a preclinical model by interrupting the inflammatory cascade induced by asbestos may in the near future change this situation [113].

## 6. The Sibaté Study: Some Lessons Learned from the Societal and Institutional Response to the Discovery of Communities Affected by Asbestos

In 1942, an asbestos-cement facility was founded in Sibaté, located 25 km from Bogotá, the capital of Colombia [88]. To investigate the complaints of inhabitants of Sibaté regarding the perceived large number of people diagnosed with asbestos-related diseases, an international scientific partnership was established in 2015. Participants included the Universidad de Los Andes from Colombia, Istituto Superiore di Sanità from Italy, the French Institute of Research for Development, Fundación Santa Fé de Bogotá (Colombia), and the Italian Universities of Bologna, Eastern Piedmont and Roma-La Sapienza. These institutions have been investigating both the health and environmental situation in the region [88]. 

The main findings of the Sibaté study are so far as follows: (1) There is a cluster of pleural mesothelioma in Sibaté, with both female and male cases. (2) Most of the people diagnosed with mesothelioma did not work in the asbestos-cement facility, which suggests the existence of non-occupational sources of asbestos exposure in the municipality. (3) Many mesothelioma patients were diagnosed very young (i.e., between 38 and 59 years-old), suggesting, based on the latency period of the disease, that they may have been exposed to asbestos during childhood [114]. (4) Landfill zones were built in the urban area of Sibaté, and in several sites of these areas, an underground layer of friable asbestos was discovered (in some cases very close to the surface), indicating the disposal of asbestos residues with a potential risk of asbestos exposure of the general population [88,115,116].

Although the first findings regarding the public health crisis in Sibaté were published in a peer-reviewed journal in April 2019 [88], and there was wide media coverage of the situation in newspapers and television news programs, after a few weeks, the case was forgotten, and no interventions were implemented to manage the identified risk.

In this context, it is important to compare the development of the asbestos-related situation in Casale Monferrato and Sibaté, which results in important lessons learned for similar public health crises in the future. In both cities, an asbestos-cement facility operated for 78 years although there are still major information gaps in Sibaté to determine if the size of the operation was similar, recognizing that the Sibaté facility was the largest of Colombia. Currently, both cities have similar population sizes (Casale Monferrato, 34,246 inhabitants [27]; Sibaté, 40,535 inhabitants [115]). In both cities, asbestos tailings were given to the population for free [10,88] although this practice in Sibaté has not been completely characterized [88]. In the early 1980s, there were rumors in Casale Monferrato of a high number of pleural cancers observed by local physicians [15]. In the case of Sibaté, these rumors came from the local population [88]. The findings of a specialization thesis in pneumology combined with the warning messages given by respected experts in the asbestos problem about the situation in 1983 resulted in the support of the mayor of Casale Monferrato and of the health secretary of the Piedmont Region to develop more in-depth epidemiological studies on the problem [15]. In contrast, the Sibaté study has barely received financial support from either health or environmental authorities at the local, regional, or national level. Moreover, although both health and environmental authorities were warned about the delicate situation, neither risk management nor remediation plans have been elaborated, and more detailed epidemiological studies have not yet been initiated in the region. Furthermore, the response of the office of the mayor of Sibaté when the cluster and contaminated zones were discovered was to hire, a few months after the first results of the Sibaté study were published, a private consulting firm that did not participate in the study to analyze in more detail the presence of the friable asbestos layer in the municipality. The results of this consulting work were not made public.

The mesothelioma cluster in Sibaté was documented initially with 15 confirmed cases [88], while the number of cases identified in Casale Monferrato that were hospitalized in the decade preceding 1983 was 70 [15]. However, in comparative terms, the number of cases in Sibaté could be misleading since major problems were identified in the official health databases that could lead to an underestimation of cases. Moreover, the age-adjusted incidence rates in Sibaté clearly indicate an excess number of mesothelioma cases in both males and, more strikingly, in females [88].

The alliance created in Casale Monferrato between academy, local physician, and both municipal and regional authorities was fundamental for the swift response to the problem that was discovered [102]. In Sibaté, although the researchers made several requests, it was impossible to establish collaborations with local authorities to address the worrisome situation that was discovered. This lack of collaboration with the municipality seems to be changing with the new administration that took office in 2020.

In terms of the population response to the problem, despite the fact that some victims in Sibaté have been very active in trying to understand and disseminate the message that there is a serious problem, these efforts have been led by a small number of persons. On the other hand, the response in Casale Monferrato included a large number of residents, and the workers union was also involved [102].

Finally, the experience of Casale Monferrato has shown that apart from the improvement of understanding the burden of asbestos-related diseases, once an excess number of cases has been demonstrated, the focus must be on remediation activities, on proper access to health care and treatment for asbestos victims, and on giving to the victims the opportunity to seek justice [102]. None of these had happened so far in Sibaté.

## 7. Key Points and Recommendations

Asbestos has long term-effects on the population’s health that may be observed in full only after the cessation of exposure, particularly because of the long latency of mesothelioma.

A mesothelioma registry is a very effective tool to monitor the negative effects of asbestos use, both at the national and local levels, given the high proportion of mesothelioma cases attributable to asbestos exposure;It takes time and perseverance to implement a mesothelioma registry. Construction is gradual and starts from the local level, prioritizing the regions where asbestos facilities or mines have operated and where the health systems are the strongest and more developed. Successful local experiences should be replicated and improve the initiatives in other regions;A mesothelioma registry needs dedicated staff. Activity must be conducted in close coordination with health centers that diagnose and treat mesothelioma cases, also in other regions;A mesothelioma registry allows to reach the following steps:Identification of cases who have been occupationally, para-occupationally, and/or environmentally exposed to asbestos;Identification of sources of exposure and of occupations where asbestos exposure occurred, providing the basis for increasing awareness of asbestos exposure and related risks;Increasing the chances of early diagnosis of mesothelioma thanks to the dissemination of information on mesothelioma;Supporting the provision of early care and treatment of people diagnosed with mesothelioma thanks to the dissemination of information on mesothelioma;Supporting the affected people demanding compensation and legal recognition of the suffered damage.Thus, the strategies to address the asbestos health effects must be carefully designed, and long-term efforts must be made to assure the flow of human capital and financial resources;Cities and towns where asbestos-cement facilities or other industrial facilities using large amounts of asbestos operated are at risk of facing a high number of cases of asbestos-related diseases, especially mesothelioma. Exposure sources include both territorial extensions contaminated with asbestos residues (i.e., brownfields) and domestic sources in houses and dwellings, such as asbestos-containing products, friable and non-friable, and materials used in the production process contaminated with asbestos;Mesothelioma clusters may first be documented with the observation of a few cases only. However, to minimize or ignore the situation because mesothelioma is a rare disease is an enormous mistake: experience has shown that in regions where asbestos exposures have occurred, the number of asbestos-related disease cases could dramatically increase over time;Consequently, after the observation of mesothelioma cases, both sporadic or clusters, it is crucial to determine if there are potential asbestos exposure sources, evaluate the risk for the population, and implement remediation and risk management plans. A national strategy for contaminated sites, including plans for both identification, remediation, and communication, is mandatory;When there is evidence of asbestos exposure for the general population or workers or evidence of asbestos-related diseases, it is strongly recommended to also set up analytical epidemiological studies. Priorities include cohorts of current and former workers of the facility, also including workers from sub-contractors that provided different services, such as cleaning, transportation, and security, and cohorts of family members and/or people that lived with the workers of the facility. Case control studies should also be considered for providing information on a broad array of possible exposure sources. Epidemiological studies should interact with the local and regional mesothelioma registries;A community at risk may be determined by the location of the homes of the workers that worked at the facility, of the workers of the facility that were hired through third parties or by the geographic areas where asbestos-containing materials were distributed for free or at discount prices, or where asbestos residues may have been placed;The use of geographic information system (GIS) data analysis has been of great value in understanding the association of asbestos exposure and mortality and morbidity in the area of influence of asbestos-using facilities: the feasibility of conducting this type of analysis should therefore be considered;When clusters of asbestos-related diseases and/or asbestos-contaminated sites are discovered, policy makers and local governments must be promptly informed. The same is recommended for results of subsequent studies and of mesothelioma registries reports;The communication of scientific findings should aim to increase awareness in the social communities about the risks from asbestos exposure and the resulting diseases;Communication should have a multidisciplinary approach, including health professionals, environmental and social scientists, affected communities and local authorities, and victims’ organizations and trade unions: the whole process should be planned and pursued with a long-lasting strategy, including periodic updates;A good communication strategy is of great importance for asbestos-related disease patients and their families considering the emotional and psychological impact of these diseases;Notwithstanding the magnitude of the problems discovered and documented, researchers and interested subjects should be prepared to face delayed response from local and national governments, the local communities, and the media: to achieve attention and an appropriate response to the problem requires perseverance in communicating the findings;Environment professionals should be trained in addressing asbestos sources and asbestos contamination as well as health professionals in identifying asbestos-related diseases to be incorporated in regional and national surveillance systems such as mesothelioma registries;The establishment of international collaborations with experienced researchers and regions with contaminated sites can be of great importance, especially in low- and middle-income countries that have not yet encountered this type of problem and need to form or expand their scientific capabilities.

## 8. Conclusions

The awareness of asbestos health effects and of the vast asbestos use worldwide urges systematic monitoring and adoption of measures to contrast the risk and to reach a definitive ban of asbestos [117]. Actions should be taken in identifying and quantifying the amount of friable and non-friable asbestos currently present in the general environment and for promoting safe asbestos removal [118].

Environmental asbestos exposure from industrial activity has affected a large number of communities, with the most relevant studies, besides those presented in this review, carried out in South Africa [119], the UK [120,121], Australia [122], Japan [123], Italy [124], the Netherlands [125], Denmark [126], Canada [127], and the USA [128]. Environmental exposure to asbestos-like fibers is also of interest from the point of view of remediation and information to the population, with relevant examples from Italy [129] and Turkey [130]. However, the communities studied and reported in the international literature are a tiny fraction of the number of communities interested in asbestos exposure and only represent the top of an iceberg. Public-access national repositories organized at a public institution would be a valuable resource to disseminate knowledge about asbestos-related risk for the general population and to support administrators seeking solutions to reduce population risk.

At the same time, the ongoing and expanding epidemiological surveillance of mesothelioma will help the retrospective identification of people exposed to asbestos as well as the control of current work conditions of workers potentially exposed to asbestos and the identification of previously unknown sources of exposure. Core items for a mesothelioma registry include the incidence estimation, the investigation of each case through an individual data collection process, the relationship with compensation issues and the periodic publishing of public-access reports, and research papers on the specific topics.

Comparing different methodologies and standardizing procedures is the first step in countries where disease surveillance systems already exist, and it is crucial to stimulate their establishment where absent.

Epidemiological surveillance systems that share scientific results with communities and stakeholders are effective tools for public health and welfare policies. They also help in ensuring exhaustiveness and equity in access to the best available therapeutic protocols for mesothelioma patients.

## Figures and Tables

**Table 1 ijerph-20-00936-t001:** Absolute numbers of incident mesothelioma cases in Casale Monferrato population broken down by sex and by period of diagnosis. Incidence in residents ever working in Eternit plant is compared to incidence in residents never working at—but indirectly exposed to—Eternit asbestos (see text for details).

	Ever Workers at Eternit: Direct Exposure to Asbestos				Never Workers at Eternit but Indirect Exposure to Eternit Asbestos				Indirect/Direct (Both Sexes)
	Women	Men	Total	M/F	Women	Men	Total	M/F	
1990–1994	9	27	36	3.0	24	20	44	0.8	1.2
1995–1999	10	24	34	2.4	38	31	69	0.8	2.0
2000–2004	11	29	40	2.6	49	59	108	1.2	2.7
2005–2009	8	21	29	2.5	51	47	98	0.9	3.4
2010–2014	2	14	16	7.0	39	53	92	1.4	5.8
2015–2017	0	7	7	-	37	29	66	0.8	9.4
Total	40	122	162	3.1	238	239	477	1.0	2.9

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
