# Peer review of "The Italian Experience in the Development of Mesothelioma Registries: A Pathway for Other Countries to Address the Negative Legacy of Asbestos"

_ijerph, 2023, doi:10.3390/ijerph20020936_

Round 1
Reviewer 1 Report
1. Is it possible to differentiate the occupational, familial and environmental exposure for Casale Monferrato case as it was done for Broni? It would enable the comparison of both cases.
2. It would be useful to define the sources of environmental exposure and adequate measures.
3. What is the difference between domestic, residential, familial, home-related, indoor asbestos exposure? It should be clearly stated as it is mentioned in different parts of the manuscript.
4. In the section concerning the review of MM registers also European countries should be included (line 521-531). MM registers are not done "only in Australia, France and South Korea" as stated in the manuscript.
5. Are there any references to confirm thesis stated in conclusions sections (line 967-971)? It is crucial due to the importance of the asbestos issue if it is intended to indicate directions directions for other countries.
6. Stating examples in references 118-125 seems not to be appropriate, seems like a random selection, e.g. 2 items from the UK and another item from Italy.
7. To address environmental exposure, asbestos removal programs would need to be put in place. While within the EU Member States are free to introduce stricter restrictions, the WHO indicates that the most effective way to eliminate asbestos-related diseases is to stop using all types of asbestos. Authors should consider indicating directions as indicated in the title of the manuscript.
Author Response
Reviewer 1:
- Is it possible to differentiate the occupational, familial and environmental exposure for Casale Monferrato case as it was done for Broni? It would enable the comparison of both cases.
Authors: We understand the interest in the reviewer’s question but in Casale Monferrato the different sources of non-occupational exposure reported by subjects very often overlap and therefore we prefer not to force a more detailed classification in this general context paper. We added the following statement in the text: For a large proportion of subjects the different sources of non occupational exposure, in particular domestic, familial and environmental showed some overlap.
- It would be useful to define the sources of environmental exposure and adequate measures.
Authors: the following sentence was added: “The sources of environmental exposure include the direct emissions from the plant, the transport of raw asbestos and of asbestos cement products to and from the factory, and the use of asbestos cement tailings in the city [10, 27, 31].”
- What is the difference between domestic, residential, familial, home-related, indoor asbestos exposure? It should be clearly stated as it is mentioned in different parts of the manuscript. >> Corrado
Authors: the following sentence was added: “Residential and environmental exposure include the exposure from external sources, the former when the exposure is linked to the location of the house; the latter has a more general meaning referring to general asbestos contamination of the environment. Domestic exposure (also defined ‘home-related’) refers to the exposure from asbestos present in the house or in tools used for everyday life. Familial exposure is used also when the exposure is linked to the occupational exposure of family members carried home with soiled work clothes [24]
- In the section concerning the review of MM registers also European countries should be included (line 521-531). MM registers are not done "only in Australia, France and South Korea" as stated in the manuscript.
Authors: We have reformulated the period, introducing some specifications about the differences among the established national epidemiological surveillance systems of mesothelioma cases by countries.
- Are there any references to confirm thesis stated in conclusions sections (line 967-971)? It is crucial due to the importance of the asbestos issue if it is intended to indicate directions directions for other countries.
Authors: The text of conclusions(rows 969-971 of first version) was modified as follows:
1) we added a reference of WHO document which states “the most efficient way to eliminate asbestos-related diseases is to stop the use of all types of asbestos.” [WHO, Elimination of asbestos-related diseases. Factsheet World Health Organization; 27 April 2014, Technical document. WHO/FWC/PHE/EPE/14.01 (https://www.who.int/publications/i/item/WHO-FWC-PHE-EPE-14.01).]
2) we added the following statement: “Actions should be taken on identifying and quantifying amount of friable and non-friable asbestos currently present in the general environment and on promoting a safe asbestos removal (EC, 2012). [European Commission. 2012. Practical guidelines for the information and training of workers involved with asbestos removal or maintenance work. https://ec.europa.eu/social/main.jsp?advSearchKey=asbestos+removal&mode=advancedSubmit&catId=22&doc_submit=&policyArea=0&policyAreaSub=0&country=0&year=0]
- Stating examples in references 118-125 seems not to be appropriate, seems like a random selection, e.g. 2 items from the UK and another item from Italy.
Authors: In our statement we intend to include the countries where, as far as to our knowledge studies have been conducted and published on the relation of occurrence of mesosothelioma with environmental asbestos exposure from industrial activities.
We have reformulated sentence and references as follow: “Environmental asbestos exposure from industrial activity has affected a large number of communities, with the most relevant studies carried out in South-Africa [118], UK [119-120], Australia [121], Japan [122], Italy [123], The Netherlands [xxx], Canada [124] US [xxx] and Turkey [125]”.
- To address environmental exposure, asbestos removal programs would need to be put in place. While within the EU Member States are free to introduce stricter restrictions, the WHO indicates that the most effective way to eliminate asbestos-related diseases is to stop using all types of asbestos. Authors should consider indicating directions as indicated in the title of the manuscript.
Authors: Authors agree that the most effective way to prevent environmental exposure is to stop using all types of asbestos and to act asbestos removal programs. Our paper aims to emphasize the role of the epidemiological surveillance of asbestos related diseases for increasing the awareness of asbestos exposure health effects and for stimulating interventions. We have added a statement at the beginning of Conclusions section, as summarized before.
Reviewer 2 Report
This study systemically described the Italian experience on the development of mesothelioma registries providing a pathway for other countries to address the negative legacy of asbestos. Registry has set the basis for further analytical studies on mesothelioma etiology including both case control analyses. In this report the authors described two Italian communities Casale Monferrato and Broni that faced an epidemic of mesothelioma resulting from the production of asbestos-cement and the diffuse environmental exposure the risk communication activities at the local and national level, with focus on international cooperation. The interaction between mesothelioma registration and medical services specialized in mesothelioma diagnosis and treatment in an area at high risk of mesothelioma.
Based on the asbestos impact on health, the authors addressed the role of the mesothelioma registry. This study provided meaningful support about the role of asbestos-associated mesothelioma. Their registry system may apply to the other countries. This manuscript is well-done and could be publishable if a minor revision could be made.
Minor points:
In Table 1. Absolute numbers of incident mesothelioma cases in Casale Monferrato population bro- ken down by sex and by period of diagnosis. Incidence in residents ever working in Eternit plant is compared to incidence in residents never working at - but indirectly exposed to Eternit asbestos.
The gender ratio for mesothelioma development is 3:1 in those who had exposed to asbestos while in those without asbestos exposure the ratio is approximately 1:1. It is most likely due to the fact that most workers are predominantly male. However, for those who had never worked at Eternit but indirect exposure to Eternit asbestos, the gender ratio remains similar. Is this also true in other regions of the world? According to our record, female patients seem to be less than male patients. Is it possible to compare with other registry databases perhaps available in Japan or Australia?
The Italian National Mesothelioma Registry (ReNaM) provides a pathway for international cooperation. It should be encouraged to share resources from different regions in the world, for instance, to set up network databases internationally. Out of curiosity, do you have set up a website to allow the readers to access to the database? It would be good if you could include a link if possible.
Author Response
Reviewer 2
This study systemically described the Italian experience on the development of mesothelioma registries providing a pathway for other countries to address the negative legacy of asbestos. Registry has set the basis for further analytical studies on mesothelioma etiology including both case control analyses. In this report the authors described two Italian communities Casale Monferrato and Broni that faced an epidemic of mesothelioma resulting from the production of asbestos-cement and the diffuse environmental exposure the risk communication activities at the local and national level, with focus on international cooperation. The interaction between mesothelioma registration and medical services specialized in mesothelioma diagnosis and treatment in an area at high risk of mesothelioma.
Based on the asbestos impact on health, the authors addressed the role of the mesothelioma registry. This study provided meaningful support about the role of asbestos-associated mesothelioma. Their registry system may apply to the other countries. This manuscript is well-done and could be publishable if a minor revision could be made.
Minor points:
In Table 1. Absolute numbers of incident mesothelioma cases in Casale Monferrato population broken down by sex and by period of diagnosis. Incidence in residents ever working in Eternit plant is compared to incidence in residents never working at - but indirectly exposed to Eternit asbestos.
Authors: We do not see a suggestion for modification. However we added the following statement: “The different gender ratio between the mesothelioma cases associated to Occupational and non-occupational asbestos exposure is noteworthy. The point will be further addresses in the following chapter on the Italian Mesothelioma Registry (ReNaM).”
The gender ratio for mesothelioma development is 3:1 in those who had exposed to asbestos while in those without asbestos exposure the ratio is approximately 1:1. It is most likely due to the fact that most workers are predominantly male. However, for those who had never worked at Eternit but indirect exposure to Eternit asbestos, the gender ratio remains similar. Is this also true in other regions of the world? According to our record, female patients seem to be less than male patients. Is it possible to compare with other registry databases perhaps available in Japan or Australia?
Authors: In the ReNaM network, a systematic analysis of gender ratio in mesothelioma incident cases has been recently performed [reference 76 in the text]. A gender ratio (M/F) equal to 2.6 has been documented in Italian caselist. In Australia 4.2 female cases have been reported for each male mesothelioma case [Australian Mesothelioma Registry. 4th report. Safe work Australia. 2015 http://www.mesothelioma- australia. com/ publications- and- data/ publications], whereas the French National Mesothelioma Surveillance Programme (PNSM) currently recording incident MM cases in 26 French geographical districts, accounting for about a quarter of the French population, provides evidence of a gender ratio equal to 3.7 [Goldberg S, Rey G, Luce D, et al. Possible effect of environmental exposure to asbestos on geographical variation in mesothelioma rates. Occup Environ Med 2010;67:417–21]. Generally, the magnitude of mesothelioma incidence in women is postivly correlated with the intensity of environmental exposure and with the dimension of female workforce in economic sector traditionally involved in asbestos exposure (mainly non-asbestos textile sector).
The statement was modified as follows: Overall, gender ratio (M/F) results equal to 2.6 in ReNaM network. Similar observations have been presente elsewhere: in Australia 4.2 female cases have been reported for each male mesothelioma case [Australian Mesothelioma Registry. 4th report. Safe work Australia. 2015 http://www.mesothelioma- australia. com/ publications- and- data/ publications], whereas the French National Mesothelioma Surveillance Programme (PNSM) currently recording incident MM cases in 26 French geographical districts, accounting for about a quarter of the French population, provides evidence of a gender ratio equal to 3.7 [Goldberg S, Rey G, Luce D, et al. Possible effect of environmental exposure to asbestos on geographical variation in mesothelioma rates. Occup Environ Med 2010;67:417–21]. However, the magnitude of mesothelioma incidence in women is positively correlated with the intensity of environmental exposure and with the dimension of female workforce in economic sectors traditionally involved in asbestos exposure (mainly non-asbestos textile sector) [76]. The analysis of mesothelioma cases in Casale Monferrato (table 1) showed a different gender ratio in relation to occupational or non-occupational exposure.
The Italian National Mesothelioma Registry (ReNaM) provides a pathway for international cooperation. It should be encouraged to share resources from different regions in the world, for instance, to set up network databases internationally. Out of curiosity, do you have set up a website to allow the readers to access to the database? It would be good if you could include a link if possible.
Authors: ReNaM national database is not directly accessible, but all ReNaM national Reports (seven editions at the present) are totally available (in Italian) at Inail website (www.inail.it). ReNaM reports include detailed tables about incidence, territorial distribution diagnosis certainty and modalities of exposure (https://www.inail.it/cs/internet/attivita/ricerca-e-tecnologia/area-salute-sul-lavoro/sorveglianza-epidemiologica-negli-ambienti-di-lavoro-e-di-vita/renam.html?id1=6443101379561#anchor). The link is included in the list of references, at number 49.
Reviewer 3 Report
Overview: article is well written and adresses an important and rare disease and shows us the whole spectrum of advantages in setting the registries. The only real disadvantaage of the article is the lenght- it is too long, everyting could be told in many less words.
Some recommendations:
11. 64-388. Chapter 2: It describes cases of 2 towns in Italy. In my opinion the chapter should be shortened, some information could be ommited as they are not important for the understanding of the situation in the town (like rows 93-95 ect).
22. Chapter 2: I also miss more subchapters to make the chapter read easily. Subchapters could be similar in Casale Monferrato case and Broni case so the reader can find similarities and disparities easily
33. 350-366 Data should be put in a Table like in a Casale Menferrato case.
44. 396-398: long latency is already well known, no need for a detailed explanation, the sentence could be omited or shortened
55. Shorten the chapter 3 and subchapters are needed (it is hard to follow otherwise)
6. 894- I agree with the fact, that spreadingthe information about mesothelioma can increase the chance for the diagnosis, but do not believe that only that information is enough for early care and treatment. Some sort of screening programes would much more affect that point.Maybe you should write line more clearly.
Author Response
Reviewer 3:
Overview: article is well written and adresses an important and rare disease and shows us the whole spectrum of advantages in setting the registries. The only real disadvantaage of the article is the lenght- it is too long, everyting could be told in many less words.
Some recommendations:
- 64-388. Chapter 2: It describes cases of 2 towns in Italy. In my opinion the chapter should be shortened, some information could be ommited as they are not important for the understanding of the situation in the town (like rows 93-95 ect).
Authors: the text was shortened, in particular the section on Broni.
- Chapter 2: I also miss more subchapters to make the chapter read easily. Subchapters could be similar in Casale Monferrato case and Broni case so the reader can find similarities and disparities easily.
Authors: the layout was harmonized. We prefer to simplify the chapter structure and to cancel the subheadings.
- 350-366 Data should be put in a Table like in a Casale Menferrato case
Authors: the chapter has been significantly changed and shortened.
- 396-398: long latency is already well known, no need for a detailed explanation, the sentence could be omited or shortened
Authors: the sentence has been deleted.
- Shorten the chapter 3 and subchapters are needed (it is hard to follow otherwise)
Authors: the chapter has been significantly shortened.
- 894- I agree with the fact, that spreading the information about mesothelioma can increase the chance for the diagnosis, but do not believe that only that information is enough for early care and treatment. Some sort of screening programes would much more affect that point.Maybe you should write line more clearly. >> Federica (hot issue the mesothelioma screening, maybe other authors could contribute)